# Obstructive Sleep Apnea, Palatal Morphology, and Aortic Dilatation in Marfan Syndrome Growing Subjects: A Retrospective Study

**DOI:** 10.3390/ijerph18063045

**Published:** 2021-03-16

**Authors:** Giuseppina Laganà, Nicolò Venza, Arianna Malara, Claudio Liguori, Paola Cozza, Calogera Pisano

**Affiliations:** 1Department of Systems Medicine, University of Rome Tor Vergata, 00133 Rome, Italy; giuseppinalagana@libero.it (G.L.); ariannam1994@libero.it (A.M.); profpaolacozza@gmail.com (P.C.); 2Sleep Medicine Centre, Department of Systems Medicine, University of Rome Tor Vergata, 00133 Rome, Italy; dott.claudioliguori@yahoo.it; 3Centre for Rare Diseases for Marfan Syndrome and Related Disorders, Department of Cardiac Surgery Unit, University of Tor Vergata General Hospital, 00133 Rome, Italy; lindapisano82@gmail.com

**Keywords:** Marfan syndrome, pediatric subjects, obstructive sleep apnea, palatal morphology, aneurysmal aorta

## Abstract

The main cause of mortality in Marfan syndrome (MS) is aortic disease. Obstructive sleep apnea (OSA) is highly prevalent in MS, and it is also associated with cardiovascular risk and maxillary deformities. The aim of this research was to analyze the possible relations between OSA, palatal morphology, and aortic root diameters in growing subjects. A group of 15 growing subjects with MS and a control group of healthy peers was selected. All of them underwent nocturnal polygraphic monitoring, digital dental casts, and transthoracic echocardiography. The results showed that OSA indexes and aortic diameters were significantly higher in the study group. Transversal palatal dimension was smaller in MS patients. Moreover, OSA severity was not related with maxillary contraction severity in MS. No correlation between OSA severity and increased aortic diameters occurred. A high prevalence of OSA, increased aortic dimensions, and significant contraction of palatal dimensions were observed in MS compared with the control group. None of these single factors may directly be associated with aortic dilation, but together, they might contribute to the development and progression of aortic aneurysm from a young age.

## 1. Introduction

Marfan syndrome is a pleiotropic inherited connective tissue disorder with a reported incidence of one in every 3000 to 5000 individuals. The disease occurs worldwide, with no preference for race or gender [1,2]. Marfan syndrome manifestations typically involve skeletal, ocular, cardiovascular, respiratory, muscular, and articular systems [3]. Among the cardiovascular manifestations, the main cause of mortality and morbidity is due to aortic disease and its complications [4].

Aortic aneurysm and dissection remain the most life-threatening manifestations of Marfan’s syndrome, and it represents the leading cause of mortality in this clinical group. Accordingly, up until now, research in Marfan syndrome has focused largely on treatment of aortic dilation and prevention of dissection [5]. Prognosis in Marfan syndrome is mainly determined by the progressive dilatation of the aorta; without surgical intervention, patients are at greater risk for aortic dissection, aortic rupture, and death at young age [6].

Within the noncardiovascular features, obstructive sleep apnea (OSA) seems to be highly prevalent in Marfan syndrome. The prevalence rate of OSA reported is 59.7% among adults (Apnea Hypopnea Index cutoff of five events/h) [7] and 80% among children (Apnea Hypopnea Index cutoff of one event/h) [8]. In Marfan syndrome patients, the underlying cause of OSA is not well elucidated but might be related to specific craniofacial features and higher upper airway collapsibility [9].

The relation between OSA and cardiovascular risk has been widely studied in the general population. Individuals with OSA seem to have a higher cardiovascular risk, showing increased prevalence of hypertension, stroke, arrhythmia, and aortic events, although these findings are not always consistent [10,11]. 

Kohler et al. [12] indicates the following factors as the main biological mechanisms underlying the causal link between OSA and arterial disease:-Intermittent hypoxia causes an increase in oxidative stress, systemic inflammation, and sympathetic stimulation;-The oscillations of the intrathoracic pressure cause an excessive mechanical stress on the arterial walls;-Reflex sympathetic activation induced by excitation associated with recurrent increases in blood pressure.

Concerning oral manifestations, Marfan syndrome may be associated with retrognathia, dolichocephaly, crowded teeth, temporomandibular joint disorders, and partial anodontia [13]. Numerous studies reveal metabolic and compositional alterations of the periodontal ligament and alteration on periodontal tissues [14]. Moreover, Marfan subjects have a specific palatal morphology, which is characterized by high palatal vault and a narrow maxillary posterior region [15]. The implication of palatal morphology in the multifactorial etiology of OSA was confirmed by previous researchers showing the relationship between maxillary deformities and increased nasal airway resistance [9,16]. 

The significant correlations between OSA and craniofacial structures and between OSA and cardiovascular disease are well described in literature; nevertheless, as far as we know, the association and the relationship between the above-mentioned variables have never been considered in Marfan syndrome growing subjects. 

Aiming to reduce the risk of aortic disease in Marfan syndrome subjects, research focused on the detection of risk factor and early risk assessment of aortic dissection. The early detection of aortic dissection risk could improve the prognosis of Marfan syndrome patients. 

The primary objective of this study was to evaluate the simultaneous presence of aortic root diameter dilatation and palatal constriction in a group of Marfan syndrome growing subjects comparing with a control group.

Furthermore, the second aim was to study the possible association between OSA, palatal morphology, and cardiovascular features and delineate the underlying mechanisms.

## 2. Materials and Methods

### 2.1. Subjects’ Selection

The study group was selected from September 2015 to December 2018 from the Centre for Rare Diseases for Marfan Syndrome and Related Disorders of Tor Vergata University Hospital. The inclusion criteria were genetic assessment of Marfan syndrome, subjects with 8–16 years of age, mixed or permanent dentition. Exclusion criteria were previous aortic root surgery, thumb-sucking habits, previous orthodontic treatment, cleft lip, and/or palate. The control group was recruited from the local population from November 2018 and January 2019. In order to compare the physical characteristics, control group subjects were selected to have values of average/frequency of age, gender, height, and weight that were as close as possible to the study group.

### 2.2. Nocturnal Polygraphic Cardiorespiratory Monitoring

All patients and controls underwent nocturnal polygraphic cardiorespiratory monitoring, which was diagnosed according to American Academy of Sleep Medicine (AASM) criteria [17]. The recording montage included an oronasal pressure cannula to record airflow, snoring sound, piezoelectric belts to detect thoracic and abdominal respiratory effort, finger pulse oximetry, heart rate, and body position sensors. The following oxygen saturation (SaO2) parameters were calculated: mean SaO2, lowest SaO2, time spent with SaO2 < 90% (T < 90), and oxygen desaturation index (ODI) (number of oxygen desaturations ≥3/h). A + H / h was defined as the sum of all apneas (>90% reduction in airflow for >10 s) and all hypopneas (>30% reduction in airflow >10 s) associated with ≥3% O2 desaturation per hour [17,18]. Digital measurements were taken of palatal volume and area. Dental casts were taken before any treatment, and the maxillary ones were scanned using the extraoral scanner OrthoXscan (Dentaurum 6mmbh E Co, Ispringen, Germany). In order to measure the palatal surface area and calculate palatal volume, the gingival plane and a distal plane were used as boundaries for the palate. The gingival plane was obtained by connecting the center of the dento-gingival junction of all erupted permanent and deciduous teeth. The distal plane was created through two points: the distal of the second deciduous molars perpendicular and the gingival plane. The same operator performed all measurements of study casts. The measurements were repeated three times, and the mean value of the palatal surface area and palatal volume were used.

### 2.3. Echocardiographic Measurements

Aortic root measurements were obtained from transthoracic echocardiograms using American Society of Echocardiography recommendations [19]. The method was similar to that used by Gautier et al., and it followed the published recommendations [20]. Aortic root diameters were measured at five levels on a parasternal long-axis view: aortic annulus; Valsalva sinus; aortic bulb; sinotubular junction; and ascending aorta. The aortic annulus was measured in systole, whereas other aortic root diameters were measured in diastole. Aortic root measurements were taken from leading edge to leading edge, parallel to the plane of the aortic valve and perpendicular to the axis of the blood flow (Figure 1). Aortic root diameter Z-scores were calculated using Gautier’s formula. Z-scores are a means of expressing the deviation of a given measurement from the size-specific population mean. Aortic root dilation was defined as any aortic root measurement Z-score ≥ 2 at the Valsalva sinus level or at the sinotubular junction or at the ascending aorta level. The growth speed of aortic diameters was defined as the ratio between the difference of measured aortic diameters and the delay, in years, between two examinations. As a result of the natural growth of the aortic root in children, a threshold of maximum growth speed above the 90th percentile (5 mm per year) was set as being clinically relevant. The difference between the first and last Valsalva sinus Z-scores available during follow-up was calculated and divided by the delay between the two measurements to obtain the Valsalva sinus Z-score evolution speed during the follow-up period [21].

### 2.4. Statistical Analysis

All statistical analyses were performed using SPSS statistical software (Statistical Package for Social Sciences, version 16.0, SPSS Inc., Chicago, IL, USA). Continuous data were expressed as means and standard deviations; categorical data were expressed as percentages and absolute values. Differences between the study and control group were compared using the t-test, Welch test, or Wilcoxon test for quantitative variables and Chi-squared test for qualitative variables. Correlation analysis in the Marfan syndrome group between palatal morphology and polygraphic cardiorespiratory monitoring index, cardiorespiratory monitoring index, and echocardiography-derived sizes were performed using Pearson’s correlation coefficient. All reported *p*-values are two-sided, and a value of *p* < 0.05 was considered statistically significant.

### 2.5. Ethical Issue

This project was approved by the Ethical Committee of Tor Vergata University Hospital (Protocol number: 4544/2017). Informed consent was obtained from all patients’ parents.

## 3. Results

The study group was composed of 15 Italian subjects with genetic assessment of Marfan syndrome, mean age of 11.27 ± 2.31 y. The control group was composed of 15 non-syndromic subjects with a mean age of 10.66 ± 2.22 y. 

Table 1 summarizes the clinical and demographic characteristics of the analyzed subjects in the two groups. In order to compare the two groups, all the subjects had similar age, sex distribution, height, weight, body mass index, and body surface area.

The echocardiography-derived sizes measured from the Marfan and control groups are reported in Table 2. Measurements showed increased aortic diameters in the study group, especially in the sections of the aortic annuls and aortic root. The z-score > 2 indicated an aortic size greater than the physiological levels in the study group. The Z-score was significantly lower in control subjects than in the Marfan subjects.

The main values obtained by the nocturnal polygraphic cardiorespiratory monitoring in the two groups are reported in the Table 3. The *t*-test showed a significant increased value in the Marfan syndrome group for the apnea hypopnea index and obstructive apnea index. The value of mean saturation of oxygen was significantly higher in the control group.

The data shown in Table 4 report the mean values of the palatal surface area and palatal volume measurements in the two groups. Palatal surface area and palatal volume were significantly reduced in the Marfan group.

The correlation analysis performed on the Marfan syndrome group did not show significant correlation between the analyzed variables, as reported in Table 5 and Table 6.

## 4. Discussion

The aim of this study was to analyze possible associations between OSA, palatal morphology, and aortic root diameters in a group of Marfan syndrome growing subjects compared to a control group. As far as we know, this is the first study providing data on the possible relationship between the mentioned variables in growing subjects with Marfan’s syndrome. Aortic disease and its complications remain the main cause of morbidity and mortality in patients with Marfan’s syndrome [4]. Thus, the prevention of aortic complications is a hotly debated topic in the literature concerning caring for patients with Marfan’s syndrome.

Our clinical research considered a group of growing subjects in order to provide useful information about the preventive measures of cardiovascular and cardiac complications.

Our results showed a high prevalence of OSA in young Marfan patients (80%) compared to the control group (13%). More in depth, the apnea hypopnea index and obstructive apnea index were higher in the Marfan subjects when compared with the control group. Moreover, oxygen saturation values were statistically significant lower in the study group, too. Existing studies reported a high prevalence of OSA in Marfan syndrome growing subjects compared with healthy subjects [9,22]. According to the literature, the early diagnosis of OSAS in growing subjects can prevent the development of neurocognitive and behavioral complications, growth retardation, systemic arterial hypertension, pulmonary hypertension, and cardiovascular and metabolic pathologies [23]. These complications can further aggravate the clinical situation of Marfan subjects.

Furthermore, our results showed a statistically significant reduction in the palatal area and in the volume of Marfan subjects compared to the control group; our data confirmed what is described in the literature regarding the reduction of the palate size in subjects affected by Marfan syndrome [1,6,9,15].

Nevertheless, in our study, no significant correlation has been observed between the severity of palatal vault contraction and the severity of OSA (evaluated by apnea hypopnea index). These findings confirm the ones obtained by Kohler et al. (2009) [24]. Differently from the group we examined, the age of subjects considered by Kohler et al. changed from 18 to 75 years. The authors described reduced intermolar distance and increased palatal height compared with control subjects, but no correlations with OSA severity, as assessed by apnea hypopnea index, were detected. Our results are consistent with a previous study carried out in 2018 on 20 growing subjects with Marfan syndrome (mean age = 8.8 years) by Paoloni et al.; the clinical group was analyzed and compared with a control group of 17 subjects (mean age = 8.5 years) [8]. The reported evidence may explain that the narrow palate may not be directly responsible for OSA syndrome in Marfan children, but it could be a co-factor.

Our echocardiography measures showed increased aortic diameters in Marfan patients compared to controls, especially in the sections of the aortic annuls and aortic root. The Z-score was significantly higher in Marfan subjects than in the control group.

According to our results, the ascending aorta dilatation is a typical feature both in adult and children with Marfan’s syndrome [4,24,25,26].

Despite polysomnographic indexes being described in the literature as clinical conditions predisposing to aortic dilatation, none of these factors were statistically linked to an increase of aortic dimensions in our study. In 2010, Kohler et al. analyzed 44 adult Marfan subjects, finding a direct correlation between the aortic diameter and the apnea hypopnea index and the maximum aortic diameter detected by echocardiography [11]. The authors considered subjects who presented other risk factors for the OSAs and cardiovascular problems development, such as advanced age and BMI. In order to evaluate these risk factors and obtain data that could provide truthful indications regarding the possible development of aneurysm and dissection of the aorta, we analyzed growing subjects with physiological BMI. The absence of direct correlations between the variables considered could be caused by the limited number of the examined sample, which can represent a limitation to our study.

In conclusion, even if a significant correlation between the OSA severity and increased aortic size was not found in this study, it can be hypothesized that the aortic dilatation in Marfan’s patients can be related to the interaction of numerous factors, including the following:-Anatomical factors (e.g., palatal dimension, retrognathia, pectus excavatum) [9,12,15,27]-Histological factors (altered composition of vascular endothelium due to the alteration of Fibrillin-1) [4,28]-Hemodynamic factors (peaks of blood pressure even higher than 80 mmHg against the walls of the vessels due to the sympathetic activation of the post apnea reflex) [29,30,31]-Biochemical factors (oxidative stress, altered repair capacity of the vessels, and vascular inflammation caused by intermittent pressure peaks on the vessel walls) [32].

None of these single factors may probably be statistically associated with aortic dilation, but together, they might contribute to the development and progression of aortic aneurysm at a young age, too.

## 5. Conclusions

The main focus of the research was to evaluate the simultaneous presence of aortic root diameter dilatation and palatal constriction in a group of Marfan syndrome growing subjects compared with a control group.

According to results of our study, the aortic dimensions of the Marfan subjects, compared with the control group, appear to be significantly increased especially in the aortic annulus and aortic bulb regions. Aortic diameter monitoring is recommended to identify patients at high risk of dissection from a young age.

Marfan subjects have sleep disorders more frequently than the control group. As a result, a careful screening carried out by polysomnography is recommended for all Marfan syndrome subjects from a very young age.

The palate morphology of Marfan syndrome subjects is characterized by a significant contraction compared to healthy subjects. The orthodontist may play a very important role within the multidisciplinary team following Marfan syndrome patients.

Secondary possible associations between cardiovascular features, OSA, and palatal morphology were explored, but none of them were detected.

In conclusion, the development of aortic dilatation, according to the data collected by this study, is the result of a complex interaction of genetic, anatomical, metabolic, and functional factors. Intervening on one of these factors could allow reducing the risk of development of aortic diseases. Considering the limitations of this study, further investigations on a larger sample of Marfan growing subjects may contribute to provide new data on the relationship between the factors involved in the development of the aortic aneurysm at a young age and clearly outline the pathophysiology of this issue.

## Figures and Tables

**Figure 1 ijerph-18-03045-f001:**
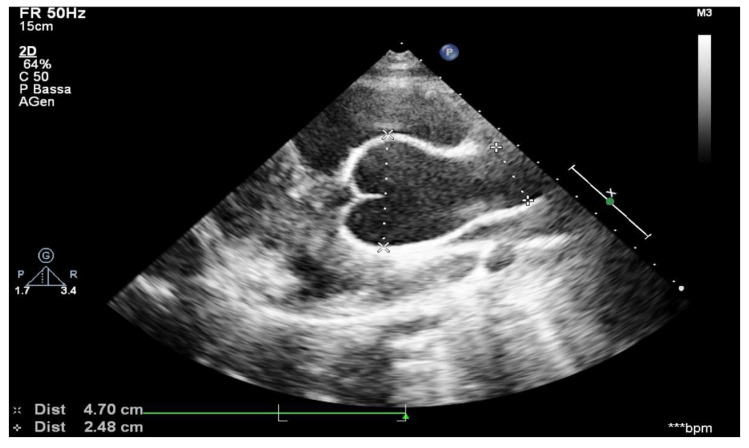
Aortic arch from suprasternal view.

**Table 1 ijerph-18-03045-t001:** Clinical and demographic characteristics in the study group and control group.

	Study Group (*n* = 15)	Control Group (*n* = 15)	*p* Value
Age	11.3 ± 2.3	10.7 ± 2.2	0.475
Sex	8 M, 7 F	8 M, 7 F	-
Height (cm)	145.7 ± 13.4	142.2 ± 10.3	0.433
Weight (Kg)	41.1 ± 9.5	45.9 ± 1.9	0.067
Body Mass Index (Kg/m^2^)	21.5 ± 3.3	22.4 ± 3.9	0.457
Body Surface Area (m^2^)	1.28 ± 0.3	1.32 ± 0.1	0.063

**Table 2 ijerph-18-03045-t002:** Echocardiography-derived sizes.

Index	Study Group	Control Group	*p*-Value
Aortic *Annulus* (mm)	19.5 ± 1.1	17.5 ± 2.8	0.015 *
Ascending Aorta (mm)	18.5 ± 2.6	20.3 ± 2.5	0.065
Aortic Bulb (mm)	25.9 ± 4.5	21.3 ± 2.9	0.003 *
Aortic Arch (mm)	19 ± 0	18 ± 0	-
Sinotubular Junction (mm)	17.6 ± 2.8	18.5 ± 2.3	0.329
Expected (mm)	21.1 ± 2.8	23.9 ± 1.0	0.002 *
Z Score	2.7 ± 2.3	−1.4 ± 1.4	<0.001 *
Ratio	1.3 ± 1.1	−0.6 ± 0.6	<0.001 *

* = *p*-value < 0.05.

**Table 3 ijerph-18-03045-t003:** Mean value of the nocturnal polygraphic cardiorespiratory monitoring.

Index	Study Group	Control Group	*p*-Value
Apnea hypopnea index/h	6.2 ± 6.7	0.5 ± 0.2	0.005 *
Obstructive apnea index/h	2.0 ± 2.9	0.4 ± 0.2	0.034 *
Central apnea index/h	0.7 ± 1.1	0.1 ± 0.2	0.090
Mixed apnea/h	0.1 ± 0.2	0.2 ± 0.2	0.398
Hypopnea/h	4.0 ± 7.1	0.4 ± 0.3	0.101
Mean saturation of the oxygen (%)	91.3 ± 2.0	96.0 ± 0.7	0.049 *
Time saturation under 90% (%)	1.9 ± 5.2	0.2 ± 0.2	0.104

* = *p*-value < 0.05.

**Table 4 ijerph-18-03045-t004:** Palatal surface area and palatal volume.

	Study Group	Control Group	*p*-Value
Palatal surface area (mm^2^)	1043 ± 115	1553 ± 133	<0.001 *
Palatal volume (mm^3^)	2920 ± 663	4381 ± 901	<0.001 *

* = *p*-value <0.05.

**Table 5 ijerph-18-03045-t005:** Correlation analysis of the Marfan syndrome group between palatal morphology and polygraphic cardiorespiratory monitoring index.

	Palatal Surface Area	*p*-Value	Palatal volume	*p*-Value
Apnea hypopnea index/h	−0.253	0.363	−0.347	0.205
Obstructive apnea index/h	0.163	0.562	0.208	0.457
Central apnea index/h	−0.342	0.212	−0.022	0.939
Mixed apnea/h	−0.236	0.397	−0.061	0.828
Hypopnea/h	−0.225	0.420	−0.401	0.139
Mean saturation of the oxygen (%)	0.279	0.315	0.206	0.461
Time saturation under 90% (%)	−0.036	0.899	0.083	0.769

**Table 6 ijerph-18-03045-t006:** Correlation analysis of the Marfan syndrome group between palatal morphology and cardiorespiratory monitoring index and echocardiography-derived sizes in the study group.

		BSA	Aortic Anulus	Ascending Aorta	Aortic Bulb	STj	Expected	Z Score	Ratio
Palatal surface area	correlation	−0.05	−0.32	−0.04	−0.20	−0.05	−0.05	−0.18	−0.14
	*p*-value	0.865	0.239	0.888	0.471	0.859	0.865	0.511	0.624
Palatal volume	correlation	−0.084	−0.409	0.046	−0.227	0.022	−0.084	−0.187	−0.167
	*p*-value	0.765	0.130	0.870	0.416	0.939	0.765	0.504	0.551
Apnea hypopnea index	correlation	−0.25	−0.18	−0.12	0.09	0.00	−0.25	0.26	0.33
	*p*-value	0.373	0.532	0.673	0.741	0.992	0.373	0.343	0.236
Obstructive apnea index	correlation	−0.20	−0.15	0.24	0.25	0.14	−0.20	0.40	0.42
	*p*-value	0.467	0.595	0.391	0.365	0.607	0.467	0.135	0.118
Central apnea index	correlation	−0.20	−0.35	−0.28	−0.34	−0.03	−0.20	−0.23	−0.25
	*p*-value	0.469	0.206	0.309	0.219	0.914	0.469	0.415	0.373
Mixed apnea	correlation	0.25	0.24	0.16	0.18	0.39	0.25	0.03	−0.04
	*p*-value	0.372	0.394	0.566	0.515	0.154	0.372	0.911	0.897
Hypopnea	correlation	−0.11	−0.01	−0.07	0.17	0.02	−0.11	0.25	0.31
	*p*-value	0.710	0.983	0.795	0.537	0.933	0.710	0.360	0.266
Mean saturation of the oxygen	correlation	0.01	0.03	0.07	−0.05	−0.31	0.01	−0.06	−0.07
	*p*-value	0.965	0.907	0.816	0.865	0.265	0.965	0.833	0.817
Time saturation under 90%	correlation	−0.19	−0.22	−0.01	−0.02	0.39	−0.19	0.11	0.14
	*p*-value	0.493	0.421	0.976	0.945	0.150	0.493	0.707	0.628

BSA = Body Surface Area, STj = sinotubular junction

## Data Availability

The datasets used and/or analyzed during the current study are available for any reasonable requests for data should be sent to the corresponding author who will review the request and give permissions.

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
