# Peer review of "Obstructive Sleep Apnea, Palatal Morphology, and Aortic Dilatation in Marfan Syndrome Growing Subjects: A Retrospective Study"

_ijerph, 2021, doi:10.3390/ijerph18063045_

Round 1
Reviewer 1 Report
In this work, Dr.Lagana with colleagues studied the association between Obstructive Sleep Apnea and Marfan syndrome. This work is similar to their other study, but this time they attempted to find a correlation between the severity of the OSA and the aortic dilation. While they did not find a significant correlation, the authors provided a thorough examination and I find this study being important for the field. However, there are some aspects which need to be corrected before the publication.
In the Discussion, authors should discuss their own results and possible connections with other studies, not provide a review of other people works. Factors indicated by Kohler can be presented in the Introduction but not in the Discussion.
The authors should discuss possible reasons why they did not detect the correlation, was the sample size too small?
Lines 195-202, which study results are discussed here? Why are they called preliminary? Please, also provide the reference for the 2018 study here.
Line 208: This study investigated correlation but not causation therefore cannot provide conclusions about etiology in this context.
Lines 250-254: this work does not provide data for such a conclusion. What do the authors mean by “Further investigations are needed to clearly outline the pathophysiology of this issue.” Please elaborate.
Minor comments:
Line 145: “Table 1 summarizes” (not resumes)
Please provide units in tables where applicable.
Line 181: “A + H index” – previously authors always used full name, it is preferable to keep it that way for consistency.
Author Response
In this work, Dr.Lagana with colleagues studied the association between Obstructive Sleep Apnea and Marfan syndrome. This work is similar to their other study, but this time they attempted to find a correlation between the severity of the OSA and the aortic dilation. While they did not find a significant correlation, the authors provided a thorough examination and I find this study being important for the field. However, there are some aspects which need to be corrected before the publication.
In the Discussion, authors should discuss their own results and possible connections with other studies, not provide a review of other people works. Factors indicated by Kohler can be presented in the Introduction but not in the Discussion.
The Discussion section was rewritten as you suggested
The authors should discuss possible reasons why they did not detect the correlation, was the sample size too small?
The possible reasons we did not detect correlations was better explained in the Discussion section as you indicated
Lines 195-202, which study results are discussed here? Why are they called preliminary? Please, also provide the reference for the 2018 study here.
The sentence was better explained in the discussion section. The cited reference was Paoloni V, Lombardo C, Placidi F, Ruvolo G, Cozza P, Lagana G. Obstructive sleep apnea in children with Marfan syndrome: relationships between three dimensional palatal morphology and apnea-hypopnea index. Int J Pediatr Otorhinolaryngol.;112:6–9 (2018).
Line 208: This study investigated correlation but not causation therefore cannot provide conclusions about etiology in this context.
The sentence was deleted and the concept was better explained in the discussion section
Lines 250-254: this work does not provide data for such a conclusion. What do the authors mean by “Further investigations are needed to clearly outline the pathophysiology of this issue.” Please elaborate.
The sentence was deleted and the concept was better explained in the conclusions section
Minor comments:
Line 145: “Table 1 summarizes” (not resumes)
Line 145 was corrected as you suggested
Please provide units in tables where applicable.
Units were added in the Tables 1, 2, 3, 4, 5.
Line 181: “A + H index” – previously authors always used full name, it is preferable to keep it that way for consistency.
Line 181 was corrected as you suggested
Please see the attachment

Reviewer 2 Report
Lagana et al. analyzed data from Marfan syndrome subjects to assess possible associations between obstructive sleep apnea (OSA), palatal morphology and aortic root diameters. They observed a high prevalence of OSA, increased aortic dimensions and significant contraction of palatal dimensions in the patients compared to controls, but did not observe a correlation between palatal contraction and severity of OSA, or between aortic size and OSA severity. This is an interesting study, but the following will need to be addressed first in order to make the study more coherent.
- The authors should describe more clearly what the motivation behind their aim is and how/why it would be beneficial to Marfan syndrome subjects to assess features including palatal morphology, aortic root diameters and OSA severity.
- The p-values in many of the tables are comma-separated and needs to be corrected.
- In Table 1, the p-value for body surface area is .036, does this mean there was a significant difference in this feature between the study and control group? The authors should include that in the manuscript if so.
- Table columns, especially in Table 6 need to clearly described - for e.g., what is BSA Dubois, Ascending aorta, what is aortic bulb and what is its unit, what is “expected”? Similarly, please define what the “/h” that appears multiple times mean (per hour?).
- The authors observe that Apnea Hypopnea Index and Obstructive Apnea Index were higher in the Marfan subjects, and also the oxygen saturation values were statistically significant lower in the Marfan subjects - it would be useful to discuss a little more in detail as to what the implications of such differences could be and what it means.
- The authors talk about “aortic dilatation” a number of times in the manuscript, but it is not clear how this is defined - is it based on a cut-off value for the aortic diameter and if so, what is the cut-off value? This should be clarified in the manuscript.
- There are several grammatical errors and non-coherent sentences, especially in the discussion and conclusion sections which needs to be corrected to make the manuscript more readable.
Author Response
Lagana et al. analyzed data from Marfan syndrome subjects to assess possible associations between obstructive sleep apnea (OSA), palatal morphology and aortic root diameters. They observed a high prevalence of OSA, increased aortic dimensions and significant contraction of palatal dimensions in the patients compared to controls, but did not observe a correlation between palatal contraction and severity of OSA, or between aortic size and OSA severity. This is an interesting study, but the following will need to be addressed first in order to make the study more coherent.
- The authors should describe more clearly what the motivation behind their aim is and how/why it would be beneficial to Marfan syndrome subjects to assess features including palatal morphology, aortic root diameters and OSA severity.
The motivation behind the aim and the clinical interest of the research was better explained in the discussion section as you suggested.
- The p-values in many of the tables are comma-separated and needs to be corrected.
P-values were corrected in the tables.
- In Table 1, the p-value for body surface area is .036, does this mean there was a significant difference in this feature between the study and control group? The authors should include that in the manuscript if so.
Coherently with the other values ​​of the physical characteristics of the subjects also the value of the body surface area was not significant. This is a typing error. The correct value is .063
- Table columns, especially in Table 6 need to clearly described - for e.g., what is BSA Dubois, Ascending aorta, what is aortic bulb and what is its unit, what is “expected”? Similarly, please define what the “/h” that appears multiple times mean (per hour?).
“BSA Dubois” represent the Body Surface Area calculated with the Dubois method. “Ascending Aorta”, “Aortic bulb” etc. were the diameters of the aorta evaluated at various sections. “/h”, associated with polysomnographic indices, means per hour of sleep. The uints were better explained in tables 1, 2, 3, 4, 5. Table 6 summarize the correlations and the units were removed.
- The authors observe that Apnea Hypopnea Index and Obstructive Apnea Index were higher in the Marfan subjects, and also the oxygen saturation values were statistically significant lower in the Marfan subjects - it would be useful to discuss a little more in detail as to what the implications of such differences could be and what it means.
This data were better discussed was clearly explained in the discussion section as you suggested.
- The authors talk about “aortic dilatation” a number of times in the manuscript, but it is not clear how this is defined - is it based on a cut-off value for the aortic diameter and if so, what is the cut-off value? This should be clarified in the manuscript.
Aortic root dilation was defined as any aortic root measurement Z-score ≥ 2 as described in the Material and Methods section.
- There are several grammatical errors and non-coherent sentences, especially in the discussion and conclusion sections which needs to be corrected to make the manuscript more readable.
The manuscript was revised by a native English reviewer.